# Multifaceted Elevation of ROS Generation for Effective Cancer Suppression

**DOI:** 10.3390/nano12183150

**Published:** 2022-09-11

**Authors:** Huizhe Wang, Mengyuan Cui, Yanqi Xu, Tianguang Liu, Yueqing Gu, Peng Wang, Hui Tang

**Affiliations:** 1Stem Cell Clinical Research Center, Shandong Provincial Hospital Affiliated to Shandong First Medical University, Jinan 250021, China; 2Department of Biomedical Engineering, School of Engineering, China Pharmaceutical University, Nanjing 210009, China

**Keywords:** lactate oxidase, ROS, PDT, CDT, cancer

## Abstract

The in situ lactate oxidase (LOx) catalysis is highly efficient in reducing oxygen to H_2_O_2_ due to the abundant lactate substrate in the hypoxia tumor microenvironment. Dynamic therapy, including chemodynamic therapy (CDT), photodynamic therapy (PDT), and enzyme dynamic therapy (EDT), could generate reactive oxygen species (ROS) including ·OH and ^1^O_2_ through the disproportionate or cascade biocatalytic reaction of H_2_O_2_ in the tumor region. Here, we demonstrate a ROS-based tumor therapy by integrating LOx and the antiglycolytic drug Mito-LND into Fe_3_O_4_/g-C_3_N_4_ nanoparticles coated with CaCO_3_ (denoted as **FGLMC**). The LOx can catalyze endogenous lactate to produce H_2_O_2_, which decomposes cascades into ·OH and ^1^O_2_ through Fenton reaction-induced CDT and photo-triggered PDT. Meanwhile, the released Mito-LND contributes to metabolic therapy by cutting off the source of lactate and increasing ROS generation in mitochondria for further improvement in CDT and PDT. The results showed that the **FGLMC** nanoplatform can multifacetedly elevate ROS generation and cause fatal damage to cancer cells, leading to effective cancer suppression. This multidirectional ROS regulation strategy has therapeutic potential for different types of tumors.

## 1. Introduction

Reactive oxygen species (ROS) are active derivatives of oxygen metabolism in the microenvironment of all biological systems [1,2,3,4]. They act as a second messenger in cell signaling and are closely associated with various diseases, including cancers. It has been long postulated that cancer cells exhibit persistently high ROS levels as a consequence of genetic, metabolic, and microenvironment-associated alterations. The escalated ROS generation in cancer cells serves as an endogenous source of DNA-damaging agents that promote genetic instability and the development of drug resistance [5,6,7,8]. Although there are some negative effects of increased ROS in tumor cells, the biochemical features of ROS make it possible to explore non-surgical therapeutic strategies to kill cancer cells through ROS-mediated mechanisms.

Non-surgical therapeutic approaches based on ROS production for cancers have been developed, including chemodynamic therapy (CDT), radiotherapy, photodynamic therapy (PDT), and enzyme dynamic therapy (EDT) [9,10,11,12,13]. The CDT based on Fenton or Fenton-like reaction has focused on utilizing inorganic nanoparticles as nanoenzymes, which can produce the most toxic ROS (·OH) through the disproportionate reaction of H_2_O_2_ without external instruments [14,15,16,17]. Additionally, photodynamic therapy (PDT) relies on the light activation of photosensitizers to generate cytotoxic singlet oxygen (^1^O_2_) and induces cancer cell death through apoptosis or necrosis [18,19,20,21]. Furthermore, the enzyme dynamic therapy can effectively convert the endogenous ROS (˙O_2_^−^ and H_2_O_2_) into highly reactive ^1^O_2_ by the cascade biocatalytic reaction of loaded superoxide dismutase and chloroperoxidase responsively in the tumor region [22].

There are one or more enzymes (e.g., oxidase, catalase, and peroxidase) participating in the H_2_O_2_ metabolism. Among them, oxidase can oxidize substrates and reduce oxygen to H_2_O_2_. Different from other oxidases, the in situ lactate oxidase (LOx) catalysis display high efficiency due to the abundant lactate substrate in the hypoxia tumor microenvironment (TME) and the ultralow O_2_ reaction threshold [23,24,25]. The extracellular level of lactate in the TME can be 20-fold higher than that under normal physiological conditions [26,27]. Recently, the strategy of consuming lactate into H_2_O_2_ in the TME by lactate oxidase has emerged. The lactate oxidase was loaded in well-designed nanomaterials and carried to accumulate in tumor tissues [25,28,29]. Despite this approach playing important roles in lactate degradation, it failed to utilize the H_2_O_2_ to further oxidize a variety of toxic substrates and decrease the production of lactate, which can be transformed from pyruvate at the end of glycolysis. Hence, the strategy of elevating ROS generation and lactate exhaustion might have a profound antitumor effect.

Herein, we designed and developed a pH-responsive nanoplatform to multifacetedly elevate ROS generation by integrating LOx and the antiglycolytic drug Mito-LND into Fe_3_O_4_/g-C_3_N_4_ nanoparticles coated with CaCO_3_ (denoted as **FGLMC**). The mechanisms of which are as follows (Figure 1): (1) LOx consumes lactic acid in tumor cells, while oxidizing lactic acid to H_2_O_2_, which was further converted to ·OH by Fe_3_O_4_ NPs through Fenton-like reaction; (2) g-C_3_N_4_ combined with O_2_ under 660 nm activation can generate ^1^O_2_ to initiate PDT; (3) The antiglycolytic drug Mito-LND, which was obtained by connecting lonidamine (LND) and triphenylphosphine cation (TPP^+^) through a long alkyl chain, can accumulate and generate ROS in the mitochondria of tumor cells to cause mitochondrial dysfunction for further dynamic therapy enhancement. This nanoparticle platform based on the multifaceted elevation of ROS generation could achieve the purpose of efficient cancer treatment and has the potential to realize multiple functions of biomedical ROS regulation as a safe and efficient treatment strategy for ROS-related tumors.

## 2. Results and Discussion

### 2.1. Synthesis and Characterization of **FGLMC**

The synthetic procedure for the nanoplatform **FGLMC** (Fe_3_O_4_-LOx-MLND@g-C_3_N_4_@CaCO_3_) involved two main steps. Firstly, the Lonidamine derivative (MLND) with mitochondrial targeting was obtained from Lonidamine and (2-aminoethyl) triphenylphosphonium bromide through amide condensation (Appendix A). The structure of **FGLMC** was confirmed by ^1^H-NMR, ^13^C-NMR, and electrospray ionization mass spectrometry (ESI-MS) (Appendix A). There were 22 aromatic hydrogen signals in the range *δ* 7.23–8.20 ppm, which was consistent with the five benzene ring skeletons of MLND. The NH signal at 8.93 ppm allowed us to confirm the successful construction of the amide bond. The single peak at 5.69 ppm was the result of the coupling of the methylene group with its adjacent N atom. The positions at 4.13 and 4.02 ppm corresponded to two adjacent methylene groups, respectively. Secondly, the Fe_3_O_4_@g-C_3_N_4_-LOx-MLND (**FGLM**) nanosystem was constructed by loading MLND, Fe_3_O_4_ nanoparticles, and LOx-NH_2_ on g-C_3_N_4_, which possess a sheet structure. Specifically, MLND and Fe_3_O_4_ nanoparticles, g-C_3_N_4,_ and LOx-NH_2_ were sonicated together, placed in the dark, and then lyophilized to remove the solvent. Then, the CaCl_2_ powder was dissolved in an aqueous solution of **FGLM** and saturated with carbon dioxide, volatilized by ammonium bicarbonate in a sealed environment to obtain the final nanoplatform **FGLMC** encapsulated by CaCO_3_.

Transmission electron microscopy (TEM) (Figure 1A) and dynamic light scattering (DLS) (Figure 1B) determination indicated that Fe_3_O_4_ NPs presented regular squares and the diameter in an aqueous solution was about 5 nm. The g-C_3_N_4_ nanosheets had a sheet-like and porous structure with a diameter of 65 nm. In addition, it can be observed that the g-C_3_N_4_ nanosheets were loaded with uniform-sized Fe_3_O_4_ NPs and the diameter was 95 nm and the hydrated particle size was 115.4 nm. The surface of **FGLMC** was covered with a calcium carbonate film, and its hydrated particle size was 130 nm. In addition, the TEM-Element Mapping image of **FGLMC** indicated the presence of Ca elements, which further proved the successful loading of CaCO_3_ (Figure 1C). The ultraviolet-visible absorption spectroscopy revealed the successful encapsulation of MLND in **FGLMC** (Figure 1D).

#### 2.1.1. LOx Loading

The successful loading of LOx was verified by sodium dodecyl sulfate-polyacrylamide gel electrophoresis (SDS-PAGE) and Coomassie Brilliant Blue Staining. It showed that the **FGLMC** group and the LOx group were on the same line, indicating that LOx was widely retained in the **FGLMC** nano-drug delivery system (Figure 1E).

#### 2.1.2. Mitochondrial Targeting

We further explored the mitochondrial targeting capabilities of different components by using the JC-1 mitochondrial membrane potential probe. After the MCF-7 cells were stained and performed by flow cytometry, we found that the **FGLM** and **FGLMC** groups showed superior membrane potential to the control group (Figure 1F). 

#### 2.1.3. Ca^2+^ Releasing

The fluorescence intensity and semi-quantitative results of the calcium ion indicator probe Rhod-2AM co-incubated with MCF-7 cells were used as the standard for Ca^2+^ release in the CaCO_3_ surface layer of **FGLMC**. First of all, the Ca^2+^ release of **FGLM** and different concentrations of **FGLMC** after co-incubation with MCF-7 cells were evaluated. The results showed that, compared with the control group (PBS treatment), there was no fluorescence signal in the cells after **FGLM** (10 μg mL^−1^) was incubated with MCF-7 cells for 2 h, indicating that there was no Ca^2+^ release in **FGLM**. In contrast, the fluorescence of Rhod-2AM grew stronger as the concentration of **FGLMC** increased. Furthermore, the fluorescence was the strongest when **FGLMC** had a concentration of 20 μg mL^−1^, which emphasized the successful coating of CaCO_3_ in **FGLMC** (Figure 2A). Additionally, the linear relationship between the fluorescence intensity and the concentration of **FGLMC** was also consistent with the semi-quantitative analysis (Figure 2B), which previously implied that the CaCO_3_ coating in **FGLMC** can cause tumor cell damage through Ca^2+^ overload. Next, the sensitivity of the CaCO_3_ coating to different pH environments was studied. It demonstrated that the fluorescence intensity of Rhod-2AM in pH 5.0 was higher than pH 7.4 (Figure 2C), and the semi-quantitative data also supported the above results (Figure 2D). Based on the above analysis, we concluded that **FGLMC** can respond and release Ca^2+^ to play a vital role in the acidic environment of tumors. The relationship between the release of Ca^2+^ in **FGLMC** and the incubation time was further detected. It was found that the fluorescence intensity of Rhod-2AM ameliorated with the extension of the incubation time of **FGLMC** and MCF-7 cells and reached the maximum Ca^2+^ release after incubating for 3 h (Appendix A), which was also consistent with the results of semi-quantitative analysis (Figure 2E). It revealed that the incubation time of **FGLMC** and tumor cells increased appropriately can enhance the release of Ca^2+^ to effectively damage tumor cells.

### 2.2. ROS Generation

Because of the outstanding Fenton-like activity of **FGLMC** in solution, 2′,7′-dichlorofluorescin diacetate (DCFH-DA) was employed as the ROS fluorescent probe to determine intracellular ROS generation. The inverted fluorescence image indicated that MCF-7 cells treated with PBS and LOx showed negligible fluorescence, implying relatively low ROS levels. Alternatively, there was an improved fluorescence signal when **FGLMC** was added, while without laser irradiation. Notably, strong green fluorescence was observed for the treatment of **FGLMC** and 660 nm laser irradiation, which demonstrated that **FGLMC** could generate much more ROS within MCF-7 cells (Figure 3A). Semi-quantitative analysis of the fluorescence intensity was also consistent with the above results (Figure 3B). Since LOx cannot penetrate the cell membrane, its function was severely limited by adding LOx alone. On the other hand, g-C_3_N_4_ could not absorb laser light when **FGLMC** was without laser irradiation, which caused the nanoplatform to not exert its PDT function. Therefore, only when **FGLMC** was irradiated with the 660 nm laser could the PDT effect of g-C_3_N_4_ be exerted to the greatest extent, and most ROS could be produced. In addition, flow cytometry was further applied to assess the ROS generating ability of different components. Compared with the control group (only PBS treatment), **FGLMC** exhibited more ROS generation than other groups, indicating that only co-loading of Fe_3_O_4_ and LOx can generate the maximum amount of ROS (Figure 3C).

### 2.3. H_2_O_2_ Detection

**FGLMC** NPs can consume lactic acid in tumor cells and oxidize it to H_2_O_2_ for further Fenton reaction with Fe_3_O_4_ and produce hydroxyl free radicals to synergistically enhance tumor lethality. Therefore, the H_2_O_2_ detection kit can be used to reveal the H_2_O_2_ production and lactic acid consumption capacity of **FGLMC** NPs. The generation of H_2_O_2_ was enhanced with the up-regulated concentration of the LOx in **FGLMC**. Moreover, the H_2_O_2_ concentration produced by 100 μg mL^−1^ of **FGLMC** was the same as that produced by 49.6 μg mL^−1^ of LOx. The loading rate of LOx in **FGLMC** was calculated as 49.6% (Figure 3D,E). All in all, the results confirmed that **FGLMC** has a strong ability to consume lactic acid and can oxidize lactic acid to H_2_O_2_.

### 2.4. In Vitro Cytotoxicity

The methyl thiazolyl tetrazolium (MTT) assay was preliminary carried out to investigate the cytotoxicity of Fe_3_O_4_ and g-C_3_N_4_. The results showed that Fe_3_O_4_ and g-C_3_N_4_ were basically non-toxic to MCF-7 cells in the concentration range of 0–300 μg mL^−1^, indicating the safety of the two carrier materials (Appendix A).

In the absence of light irradiation (laser off), g-C_3_N_4_, FGLM, and **FGLMC** showed lower cytotoxicity at low doses. However, the all-active (laser on) three materials showed a certain degree of toxicity at high doses, especially the **FGLMC** group, which was attributed to the ROS-dependent promotion of cell death (Figure 4A,B). Under the laser-on conditions, there was a significant difference between the half-maximal inhibitory concentration (IC_50_) of g-C_3_N_4_ and **FGLMC** (*p* < 0.001) (Figure 4C). Such behavior could be explained by the fact that CDT and PDT synergistically enhanced ROS generation better than PDT individually. Furthermore, to fully prove the superiorities of **FGLMC** under 660 nm laser, CLSM images of calcein-AM and propidium iodide (PI) co-staining of dead/live cells after receiving different formulations were carried out. Compared to the control group and **FGLMC** without the laser group, FGLM and **FGLMC** with laser irradiation exhibited significant cell lethality. Furthermore, the MCF-7 cells incubated with **FGLMC** displayed the highest red fluorescence and negligible green fluorescence when exposed to the 660 nm laser irradiation, confirming the therapeutic synergistic effect of CaCO_3_ on killing cancer cells, which was consistent with the cell viability results (Figure 4D). Furthermore, cell apoptosis induced by different formulations was examined by flow cytometer after AV/PI co-staining. The **FGLM** and **FGLMC** group without a laser showed better early and late apoptosis rates than other treatments (Figure 4E). However, the apoptotic rate was significantly increased when the laser was on. It was worth mentioning that **FGLMC** induced the highest early (18.74%) and late (76.67%) apoptosis percentages (Figure 4F,G). These results jointly demonstrated that CDT, PDT, and Ca^2+^ overloading collaborated with MLND and exerted the best efficacy.

### 2.5. **FGLMC** Suppress Cancer Cell Migration and Invasion

As cell migration and invasion are critical for tumor growth and metastasis, Transwell assays and Scratch assays were performed. Microscopic images revealed that compared with the control group, the migration distance of MCF-7 cells was fundamentally unchanged after **FGLMC** treatment for 12 h. Nevertheless, the distance was relatively smaller after incubating for 24 h, and the cell spacing was significantly reduced after treatment for 48 h, which indicated that **FGLMC** can effectively inhibit cell proliferation and migration (Figure 5A). To compare the inhibitory ability of different drugs, LND, MLND, and **FGLMC** were cultured with MCF-7 cells for 24 h to observe cell spacing. Interestingly, **FGLMC** showed a cell migration inhibitory ability comparable to MLND, which was significantly different from the control group and the LND group (Figure 5B).

Western blotting was applied to further verify the effects of different formulations. Compared with the control and LND group, peroxiredoxin 3 (Prx3) in the cells treated with MLND was significantly oxidized, the degree of mitochondrial depolarization was strong, and the expression of Prx3 was reduced (Figure 5C). This can be attributed to the enhanced mitochondrial targeting ability after triphenylphosphine modification. In addition, due to the damage to mitochondria caused by Ca^2+^ overload, **FGLMC** exhibited the greatest inhibition of Prx3 expression. Moreover, a different concentration of **FGLMC** was added followed by 660 nm laser irradiation, the expression of Prx3 was suppressed, which revealed that PDT could also enhance mitochondria damaging capability (Appendix A). PINK1 is a mitochondrial-dependent protein kinase, which is located in the inner mitochondrial membrane of the cell, and its expression is generally very low. The expression of PINK1 in the MLND and **FGLMC** group was higher than that in the control and LND groups, which indicated that the treatment with the drug MLND would cause the mitochondria in the cells to depolarize and cause mitochondrial damage (Figure 5C). The quantitative analysis data of Prx3 and PINK1 indicate that mitochondrial damage is an important factor in cancer cell apoptosis (Figure 5D,E).

### 2.6. Antitumor Effect In Vivo

Studies were also carried out to examine the in vivo therapeutic efficacy of **FGLMC**. Mice bearing MCF-7 tumors were, respectively, injected with (1) PBS (25 μL); (2) LND (25 μL, 10 mg kg^−1^); (3) **FGLMC** (25 μL, 10 mg kg^−1^); (4) **FGLMC** + 660 nm laser (25 μL, 10 mg kg^−1^, 100 mW cm^−2^ for 5 min) (Figure 6A). After treatment, the tumor size was measured every other day in each group to identify the tumor growth inhibition effect. Compared with the rapid tumor growth curve of the PBS group, groups treated with LND and **FGLMC** displayed relatively slower growth rates. Furthermore, the **FGLMC** + 660 nm laser group showed significant tumor growth suppression (Figure 6B and Appendix A). The weights and images of tumors were well consistent with the measurement of tumor volume in vivo (Figure 6C,D). The body weights of mice in all groups increased, indicating that all formulations had negligible toxicity (Figure 6E). Similarly, images of hematoxylin and eosin (H&E) stain slices implied no obvious histological changes in major organs of the control and **FGLMC** + 660 nm laser group (Figure 6F). As expected, another mice treatment with **FGLMC** with 660 nm laser irradiation completely ablated the tumor during the 30 days, as well as produced the highest survival rate with notable elongation of life span compared to other groups, after which these mice were sacrificed (Figure 6G). These results were attributed to the synergistic effect of PDT and CDT to enhance the generation of ROS and the mitochondrial damage caused by chemotherapeutic and Ca^2+^ overload.

## 3. Conclusions

In summary, we successfully developed a versatile nanoparticle platform **FGLMC** based on multifaceted elevation of ROS generation by integrating LOx and the antiglycolytic drug Mito-LND into Fe_3_O_4_/g-C_3_N_4_ nanoparticles coated with CaCO_3_. The combination of dynamic therapy (CDT/PDT/EDT) synergistically enhanced ROS generation. Additionally, the antiglycolytic drug Mito-LND caused mitochondrial dysfunction and generated ROS for further dynamic therapy enhancement. Remarkably, this nanoplatform exhibited excellent ROS-generation ability and confirmed potent in vivo anticancer efficacy, as evidenced by the inhibition efficiency on MCF-7 tumor growth in a nude mice model with good biocompatibility. Briefly, the multifunctional collaborative nanoplatform provided a promising strategy for breast cancer therapy.

## Data Availability

All data generated or analyzed during this study are included in this published article and its Appendix A.

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
