# Peer review of "Multifaceted Elevation of ROS Generation for Effective Cancer Suppression"

_nanomaterials, 2022, doi:10.3390/nano12183150_

Round 1

Reviewer 1 Report

Dear Authors,

I believe that this study, proposed for publication, is very interesting and will have a inconsiderable impact in the extremely complex field of therapeutic strategies for cancer. Consequently, I recomand its publication.

Author Response

I believe that this study, proposed for publication, is very interesting and will have a inconsiderable impact in the extremely complex field of therapeutic strategies for cancer. Consequently, I recomand its publication.

Response: Thanks for your good comments. Special thanks to you for your efforts in reviewing this manuscript.

Reviewer 2 Report

Thank you for the opportunity to review this manuscript, dealing with interesting findings entitled “Multifaceted elevation of ROS generation for effective cancer suppression”. All findings are interesting, and the article includes a balanced and critical view of the findings. However, several calculations are confusing, in addition to the lack of proper explanations and low graphic image presentation. Scheme 1 is important but has very low quality. Therefore, author needs to redraw it with good quality graphics. It’s recommended to use better software to redraw it. Calcein density represents the mitochondrial mPTP opening which is direct correlation with Ca2+ intake, swelling, and mitochondrial dysfunction. All this description of the author's findings is not very limited. The author needs to describe it briefly. Furthermore, the calculation of Figure 5C does not seem relevant to the representative image. Therefore, I would like to see the whole immunoblot membrane image in addition to the calculation.

Author Response

  1. Scheme 1 is important but has very low quality. Therefore, author needs to redraw it with good quality graphics. It’s recommended to use better software to redraw it.

Response: Thanks for your good suggestion. Due to typographical issues, the picture of Scheme 1 is of lower resolution and it has been revised.

  1. Calcein density represents the mitochondrial mPTP opening which is direct correlation with Ca2+ intake, swelling, and mitochondrial dysfunction. All this description of the author's findings is not very limited. The author needs to describe it briefly.

Response: Thanks for your good suggestion. In this paper, the commercial calcium ion fluorescent probe Rhod-2AM was used to demonstrate calcium ion uptake, and mitochondrial dysfunction was verified by measuring mitochondrial membrane potential.

  1. Furthermore, the calculation of Figure 5C does not seem relevant to the representative image. Therefore, I would like to see the whole immunoblot membrane image in addition to the calculation.

Response: Thanks for your good suggestion. The quantification of Prx3/GAPDH in Figure 5c was confused with other data when plotted and has now been revised. The whole immunoblot membrane image was added in the Supporting Information.

Special thanks to you for your good comments.

Reviewer 3 Report

This paper deals with the development of a nanomaterial drug for effective cancer suppression.  The paper can be published after the following point would be revised;

     In title, used material would be pointed.  At the stage of the title, the paper would investigate the ROS generation, however, the authors would be ROS generation by nanomaterial.  So, the authors would be corrected the title appropriately for the focus of this paper.

     The characterization of FGLMC would be qualitative.  If the authors have quantitative data, the authors should describe it.  For example, g-C3N4 and LOx amount and activity.

     In Fig. 5A, some explanations of these results would be required.  It is difficult to understand how the red lines in Fig. 5A were obtained.

     The authors should discuss the nanomaterial would be ineffective for the normal cells.

Author Response

  1. In title, used material would be pointed.  At the stage of the title, the paper would investigate the ROS generation, however, the authors would be ROS generation by nanomaterial.  So, the authors would be corrected the title appropriately for the focus of this paper.

Response: Thanks for your good suggestion. Due to the complicated components of our nanomaterial, the name of this nanomaterial is not fit in the title. The readers could find the details in the Abstract.

  1. The characterization of FGLMC would be qualitative. If the authors have quantitative data, the authors should describe it.  For example, g-C3N4 and LOx amount and activity.

Response: Thanks for your good suggestion. About the contents of g-C3N4 and LOx in FGLMC, we didn't find proper ways to quantify them. We hope to address this issue in future research.

  1. In Fig. 5A, some explanations of these results would be required.  It is difficult to understand how the red lines in Fig. 5A were obtained.

Response: Thanks for your good suggestion. Compared with PBS, FGLMC significantly inhibited the growth of MCF-7 cells after incubation. To visualize intercellular distances more visually, red dashed lines are added. The results showed that FGLMC could effectively inhibit the migration and invasion of cancer cells.

  1. The authors should discuss the nanomaterial would be ineffective for the normal cells.

Response: Thanks for your good suggestion. Because of the short time of revise stage, we don’t have enough time to study the effects of FGLMC on normal cells. However, the FGLMC has slight effects on the cell viability of MCF-7 when the laser was OFF. Due to the EPR effect, FGLMC was accumulated at the tumor site. When the lase was ON, FGLMC should have been ineffective for the normal cells.

Special thanks to you for your good comments.